# Towards an In Vitro 3D Model for Photosynthetic Cancer Treatment: A Study of Microalgae and Tumor Cell Interactions

**DOI:** 10.3390/ijms232113550

**Published:** 2022-11-04

**Authors:** Christopher Holmes, Juan Varas, Sebastián San Martín, José Tomás Egaña

**Affiliations:** 1Institute for Biological and Medical Engineering, Schools of Engineering, Medicine and Biological Sciences, Pontificia Universidad Católica de Chile, Vicuña Mackenna 4860, Santiago 7821093, Chile; 2Biomedical Research Center, School of Medicine, Universidad de Valparaiso, Viña del Mar 2520000, Chile

**Keywords:** tumor, microalgae, photosynthesis, hypoxia, 3D model

## Abstract

As hypoxic tumors show resistance to several clinical treatments, photosynthetic microorganisms have been recently suggested as a promising safe alternative for oxygenating the tumor microenvironment. The relationship between organisms and the effect microalgae have on tumors is still largely unknown, evidencing the need for a simple yet representative model for studying photosynthetic tumor oxygenation in a reproducible manner. Here, we present a 3D photosynthetic tumor model composed of human melanoma cells and the microalgae *Chlamydomonas reinhardtii*, both seeded into a collagen scaffold, which allows for the simultaneous study of both cell types. This work focuses on the biocompatibility and cellular interactions of the two cell types, as well as the study of photosynthetic oxygenation of the tumor cells. It is shown that both cell types are biocompatible with one another at cell culture conditions and that a 10:1 ratio of microalgae to cells meets the metabolic requirement of the tumor cells, producing over twice the required amount of oxygen. This 3D tumor model provides an easy-to-use in vitro resource for analyzing the effects of photosynthetically produced oxygen on a tumor microenvironment, thus opening various potential research avenues.

## 1. Introduction

Tumor hypoxia is one of the main factors leading to increased malignancies, such as tumor recurrence, metastasis, resistance to treatments, and overall poor patient prognosis [1]. Hypoxia in tumors does not only lead to a more aggressive tumor cell phenotype, but also dampens immune cell recognition and blocks effective immune response from taking place [2,3]. A lack of proper oxygenation also leads to an intrinsic resistance to treatments that use reactive oxygen species (ROS) as their main vehicle for killing tumor cells, such as radio or photodynamic therapies [4]. It is therefore unsurprising that recent efforts have focused on developing ways to oxygenate the tumor microenvironment by using oxygen carriers, hyperbaric oxygenation, or implantable electronic devices [5,6,7,8].

In vitro 3D models are a staple for studying tumor growth, drug delivery, and treatment efficacy [9]. At present, there are numerous methods for recreating a 3D structure to mimic the tumor microenvironment. Among these methods, those that thoroughly recreate the extracellular matrix (ECM) are the ones that most closely resemble a real tumor, due to the intrinsic relationship between tumor cells and the ECM [10]. To implement these approaches, tumor cells are embedded into a 3D matrix composed of a biocompatible polymer, such as collagen or alginate, where they grow and develop similar to real tumors. The 3D matrix, commonly made of a natural or synthetic hydrogel, imitates tissue ECM and allows cell attachment, proliferation, migration, and 3D stratification [11]. Of these hydrogels, Matrigel is the most commonly employed for 3D tumor development, mainly due to its ease of use and widespread availability. Due to the nature of its production process, Matrigel is limited by its batch-to-batch variability and contaminants, which negatively affects the reproducibility of results [12,13]. Furthermore, Matrigel is mainly composed of proteins from the basal membrane of Engelbert–Holm–Swarm mouse sarcomas, presenting mostly type IV collagen, contrary to human tissues which are predominantly composed of type I collagen [14].

Natural collagen hydrogels are another commonly used alternative for cancer organoid production, as their composition resembles the human ECM, thus leading to the development of cell–matrix interactions which help tumor development and differentiation [15,16,17]. Collagen matrix cultures are highly consistent and reproducible if matrix fabrication and cell seeding is controlled and are easy to analyze by microscopy due to the autofluorescence of collagen fibers [11,18]. Collagen matrices usually also contain proteoglycans (PGs) and glycosaminoglycans (GAGs), which aid in the integration of cells with the ECM and in turn support various cellular processes, such as proliferation, migration, and angiogenesis [19]. Alterations to the ratio and presence of specific PGs/GAGs has been shown to promote carcinogenic phenotypes in cells and has been described as relevant for the development of some specific tumor types [20,21].

Over the last decade, photosynthesis has been proposed as an alternative source for oxygen to tissues in vitro [22] and in vivo [23]. By using photosynthetic microorganisms, such as microalgae and cyanobacteria, researchers have explored the induction of photosynthesis as a simple and controllable local source of oxygen for tissues. This approach has been evaluated in various in vitro and in vivo hypoxic models, such as wound healing [24], myocardial ischemia [25], 3D bioprinted constructs [26,27], and ex situ kidney preservation [28]. Moreover, a phase I clinical trial using photosynthetic collagen scaffolds with the microalgae *Chlamydomonas reinhardtii* was recently published [29], showing the safety of such photosynthetic approaches in humans.

Over the last few years, within the context of photosynthetic therapies, the use of photosynthetic microorganisms has been evaluated as an adjuvant treatment for hypoxic tumor oxygenation [30,31,32,33,34]. For instance, increased ROS production during radio-photodynamic therapy was achieved by coating the microalgae *Chlorella vulgaris* with erythrocyte membranes [30] or a calcium phosphate shell [31], leading to higher tumor cell death in mice. Similarly, co-encapsulation of *C. vulgaris* with perfluorocarbons in calcium-alginate hydrogels increased tumor cell death in vitro and in vivo [33] during photodynamic therapy. Moreover, it has also been shown that the cyanobacteria *Synechoccocus elongatus*, together with black phosphorus nanosheets, increases ROS production in photodynamic therapy, leading to increased tumor cell death and tumor shrinkage in mice [32]. Finally, this same cyanobacteria was also used in conjunction with sonodynamic therapy, increasing tumor cell death in vitro and in vivo [34].

Due to the promising results obtained by using photosynthesis for tumor oxygenation and treatment, the study of the interactions and functional relationships between tumor cells and photosynthetic microorganisms is a novel and highly relevant field of research with growing scientific interest. Therefore, in this work, the microalgae *C. reinhardtii* was co-cultured with a melanoma tumor cell line inside a collagen-GAG scaffold to form the 3D model, and their interaction was characterized by several means. The functional in vitro 3D model used in this study is presented as a platform to explore the relationship between these two cell types for photosynthetic cancer treatment.

## 2. Results

### 2.1. C. reinhardtii Survival under Cell Culture Conditions

As optimal culture conditions for *C. reinhardtii* significantly differ from human cells, the potential deleterious effect of human physiological conditions in microalgae was analyzed first (Figure 1). Osmolarity of microalgae culture media (TAP) and human cell culture media (RPMI) was measured, averaging 71.0 and 330.7 mOsm/L, respectively. Microalgae were then incubated in each media at 28 or 37 °C for 24 h. After incubation, microalgae morphology was analyzed and results showed that cell integrity was maintained in all conditions, with no obvious membrane rupturing. Interestingly, in TAP, the overall morphology of the cells was generally the same at different temperatures; in RPMI, it was slightly altered, with cells showing a reduction in cytoplasm content inside the cell wall, regardless of temperature (Figure 1A). Next, single cell area was quantified, with no significant observable differences among groups (Figure 1B). Moreover, cell death was measured and quantified by flow cytometry (Figure 1C,D), where results showed no significant effect of the culture media at 28 °C (mortality values of 5.8 ± 1.7% for TAP and 5.9 ± 1.9% for RPMI). In contrast, at 37 °C, the culture media had a significant effect as death ratio increased from 10.0 ± 2.5% in TAP to 18.0 ± 2.8% in RPMI. Next, the total chlorophyll content of the microalgae was measured in all culture conditions (Figure 1E). No effect of the culture media was observed at either temperature. In samples of 1 × 10^7^ microalgae, 28 °C values were 5.9 ± 0.5 µg for TAP and 6.1 ± 0.5 µg for RPMI, while 37 °C values were 4.7 ± 0.1 for TAP and 4.6 ± 0.5 µg for RPMI. Finally, the proliferation capacity of the microalgae was confirmed in all groups by seeding in TAP-Agar plates (Figure 1F). As such, although these microalgae were negatively affected by RPMI media and physiological temperatures, most microalgae are still viable after 24 h of incubation.

After showing that microalgae viability and morphology were not severely affected by RPMI or 37 °C, the next goal was to determine whether such conditions might have an impact on their metabolism by means of oxygen production and consumption rates. Here, microalgae were incubated in TAP or RPMI media at 28 or 37 °C for zero or 24 h in darkness, and their oxygen metabolism was measured (Figure 2). At time zero, results showed that oxygen consumption did not vary at the analyzed temperatures, with values of 0.35 ± 0.06 and 0.24 ± 0.01 nmol/(mL·s) for TAP at 28 °C and 37 °C, respectively. Similarly, no significant variations were determined for RPMI, with values of 0.3 ± 0.05 and 0.23 ± 0.03 nmol/(mL·s) for 28 °C and 37 °C, respectively. Similarly, oxygen production did not significantly vary among all conditions; values of 0.57 ± 0.03 and 0.58 ± 0.04 nmol/(mL·s) at 28 °C for TAP and RPMI, respectively, and 0.58 ± 0.03 and 0.55 ± 0.04 nmol/(mL·s) at 37 °C were recorded (Figure 2, upper panels). In contrast, after 24 h of incubation, significantly lower oxygen consumption was detected for both media at 37 °C, with values of 0.22 ± 0.01 and 0.27 ± 0.04 nmol/(mL·s) for TAP and RPMI, respectively, and 0.36 ± 0.02 and 0.31 ± 0.03 nmol/(mL·s) at 28 °C. Oxygen production was significantly reduced in all conditions, with values of 0.42 ± 0.02 and 0.43 ± 0.05 nmol/(mL·s) for TAP at 28 °C and 37 °C, respectively, and RPMI presented significantly lower values of 0.29 ± 0.01 and 0.31 ± 0.02 nmol/(mL·s) at 28 °C and 37 °C, respectively. In conclusion, at time zero, neither consumption not production was affected by any condition. After 24 h, lower consumption was reported at 37 °C independent of media, while production was overall lower than at time zero and was further reduced by RPMI and 37 °C (Figure 2, lower panels). These results illustrate that although oxygen metabolism was affected by RPMI and physiological temperatures, microalgae can still produce large quantities of oxygen even 24 h after incubation.

### 2.2. Biocompatibility of Microalgae and Tumor Cells in Co-Cultures

After characterizing the effect of human-like culture conditions over the microalgae, human melanoma cell line C32 and microalgae were co-cultured in 2D culture plates. First, a microscopic analysis with transmitted light was performed, showing that microalgae were often localized between the cells, leading to apparent morphological changes and detachment of C32 cells (Figure 3A, left). The microalgae were then washed out, and the total area of the plate covered by the cells was quantified (Figure 3A, right); there was a significant reduction in covered area in the co-culture (64.1 ± 1.9%) compared to the control (84.6 ± 2.3%). To further evaluate morphological changes in C32 cells, actin filaments were stained and visualized by fluorescence microscopy, and fiber distribution along the cells was quantified (Figure 3B, upper). Significant differences were detected between conditions, showing higher average actin fiber intensity along the periphery for co-cultures (0.89 ± 0.1 and 0.89 ± 0.1 AU for left and right, respectively) than control cells (0.66 ± 0.09 and 0.58 ± 0.09 AU left and right, respectively). To further analyze cell polarization, a fiber directionality map was generated (Figure 3B, lower). Quantification of fiber coherency showed a significant increase in co-cultures (coherency value of 0.18 ± 0.2 AU for the control and 0.27 ± 0.03 AU for co-cultures). These results indicate that co-cultured C32 cells became polarized, suggesting a migratory phenotype.

Afterwards, the effect of light in co-cultures was analyzed by several means (Figure 4). First, cell viability was quantified by an MTT metabolic assay (Figure 4A, left). As expected, after 24 h, control C32 cultures showed increased metabolic activity, both in dark (175.3 ± 24.0%) and light (149.7 ± 28.9%) conditions, while in co-cultures, metabolic activity did not significantly differ from time zero control, with values of 79.3 ± 7.8% and 71.2 ± 1.8% for dark and light, respectively. In order to evaluate if the steady state condition was due to increased mortality in the co-culture, cell death was measured by LDH release into the medium (Figure 4A, right). Co-culturing with microalgae induced a small but not significant increase in tumor cell death with respect to control cultures, with values of 1.8 ± 0.7% and 3.4 ± 0.7% for dark and light conditions, respectively. Within the same experimental setting, the viability of the microalgae in co-cultures was also analyzed and further quantified by flow cytometry (Figure 4B). Interestingly, in the absence of C32 cells, the percentage of dead microalgae was significantly higher (18.0 ± 5.6%) compared to dark (3.7 ± 1.7%) or illuminated co-cultures (2.8 ± 3.5%). Finally, the capacity of microalgae to proliferate in a co-culture setting was not affected in either illumination setting (Figure 4B). These results indicate that both cell types are biocompatible with each other under long-term incubation.

As microalgae and C32 cells proved to be biocompatible in co-cultures, the capability of microalgae to inhibit hypoxia in C32 cells was then analyzed (Figure 4C). For this, co-cultures were placed in a hypoxic chamber for 24 h and the levels of HIF-1α, the master protein and hallmark for the hypoxic-response, were evaluated by Western blot. In the absence of microalgae, low levels of HIF-1α were detected in normoxic C32 cells (0.7 ± 0.3), which significantly increased in hypoxia (19.9 ± 1.7) or in the presence of CoCl (54.1 ± 4.8), a chemical inductor of HIF-1α. In hypoxic co-cultures, HIF-1α levels increased in darkness (35.6 ± 14.1) but were inhibited when cultures were illuminated (3.6 ± 1.0). These results show that, in the presence of light, microalgae are able to oxygenate cell cultures under hypoxia.

### 2.3. Establishment and Characterization of the Photosynthetic 3D Tumor Model

Once biocompatibility was proven in a standard in vitro culture system, the next step was to engineer a scaffold-based 3D model that better resembles the tumor microenvironment. Here, C32 cells were injected into a collagen-GAG scaffold and a seeding efficiency of over 75% was obtained. In order to map and quantify the metabolic activity of the seeded scaffolds, MTT assays were performed (Figure 5A). At time zero, formazan crystals were concentrated near the injection site at the center of the scaffold, showing only some isolated metabolic clusters along the edge. However, after 24 and 48 h of incubation, metabolic activity spread within the scaffold, showing a more intense and homogeneous distribution of crystals. Compared to time zero, MTT quantification showed an increase of 180 ± 36% and 210 ± 30% at 24 and 48 h, respectively (Figure 5A, right). As no significant differences were observed between both times, 48 h was chosen for further characterization. Next, seeded scaffolds were cryosectioned and cell nuclei were stained (Figure 5B) for visualization under fluorescence microscopy. Cell distribution along the matrix was analyzed, and images were divided into 40 sections in order to count the total number of cells per section (Figure 5C). Overall, heat map analysis shows that cells were concentrated in the inner sections of the matrix, with a lower amount towards the periphery, regardless of the axis. Intra-axis quantifications show that although cells were present throughout the whole scaffold, significantly larger numbers of cells were present along the lower 50% of the Q axis (Q1 and Q2), closer to the silicone layer, and along the central sections of the D axis, between D4 to D7, both of which correlate with the approximate site of injection at the center of the scaffold. In order to get more details about the interactions between the seeded cells and the scaffold, laser confocal scanning microscopy (LCSM) studies were performed, showing that cells seemed to aggregate into clusters and form micro tumor-like structures within the scaffold (Figure 5D). White arrowheads show strong cell to cell interactions and cell proliferation (Figure 5D, upper right). Arrowheads present at the maximum intensity projection show that the cells were tightly clustered along the collagen sheets, showing direct attachment to the scaffold (Figure 5D, lower right).

After establishing the 3D tumor model, the next step was to incorporate photosynthetic microalgae into the scaffolds (co-cultured scaffolds). As shown by the overall green color, *C. reinhardtii* showed a homogeneous distribution after seeding, with a larger density at the seeding zone in the center of the scaffold (Figure 6A). Co-cultured scaffolds were incubated for 24 h and the viability of C32 cells was quantified by an MTT assay. Interestingly, just as previously observed before in 2D cultures, co-cultured scaffolds did not show differences in metabolic activity (84.2 ± 9.5%) from time zero control, while scaffolds seeded only with C32 cells (control) showed a significant increase (157.6 ± 6.0%) (Figure 6B). Aiming to characterize the interaction between both cell types and the scaffold, Scanning Electron Microscopy (SEM) images were taken, analyzed, and colored (Figure 6C). As shown by the arrowheads in the top and side views of the seeded scaffold, microalgae (green) seem to cluster among themselves or on top and in between C32 cells (red) within the collagen sheets of the scaffold. However, as delimitations between collagen sheets and C32 cells were difficult to correctly assess in SEM images, LCSM images were taken to complement this characterization, as fluorescence allows for better identification of each cell type. Here, results obtained by SEM were confirmed, as the chlorophyll autofluorescence of the microalgae was observed in a similar pattern, showing direct interaction with the microtumors (Figure 6D). 

Further, co-cultured scaffolds were histologically analyzed (Figure 7A). Hematoxylin/Eosin (H&E) staining supported SEM and LCSM results, with C32 cells forming micro-tumors. As indicated by the white arrowheads in the H&E panels, microalgae were visible around and in between the tumor cell clusters. Cell proliferation and apoptosis of the C32 cells in the scaffold was evaluated by Ki-67 and Caspase-3 staining, respectively. White arrowheads show positive cells for the corresponding staining. Unexpectedly, a significant increase in Ki-67 positive nuclei was observed in the presence of the microalgae (34.3 ± 3.3%) when compared to the control (16.1 ± 1.4%), while no significant differences were observed between groups for Caspase-3, with an average of 13.56 ± 1.3% positive cells in control scaffolds and 16.3 ± 2.7% positive cells in co-cultured scaffolds. Finally, control and co-cultured scaffolds were subjected to dark and light cycles to determine if the oxygen released by the presence of microalgae could supply the metabolic requirements of the tumor model (Figure 7B). Control scaffolds showed an equal consumption of oxygen independent of light exposure, with an average of 45.0 ± 7.0 pmol/(mL·s) when in darkness and an average of 39.0 ± 8.0 pmol/(mL·s) when exposed to light. On the other hand, co-cultured scaffolds in darkness showed more than double the oxygen consumption of control scaffolds, with an average of 96.0 ± 11.0 nmol/(mL·s). When exposed to light, co-cultured scaffolds showed high oxygen production, enough to supply the oxygen consumption needed by the tumor cells and produce an average oxygen excess of 127.0 ± 17.0 nmol/(mL·s). Therefore, microalgae inside the co-cultured scaffold effectively produce enough oxygen to supply the cellular environment, greatly exceeding C32 oxygen requirements.

## 3. Discussion

As there has been an increased interest in the treatment of hypoxic tumors with photosynthetic microorganisms [30,31,34], this work aims to generate a better understanding of the potential interactions between photosynthetic microalgae and tumor cells. As these cell types do not share a common environment in nature, their interaction is far from physiological. Nevertheless, several evolutionary examples support the notion that functional photosymbiotic relationships can be established between hetero and autotrophic systems [35]. Within this context, several groups have suggested that the induction of photosynthetic relationships could be established as a novel approach for the treatment of hypoxia-related pathological conditions [23].

Here, the interaction between the microalga *C. reinhardtii* and C32 melanoma cells was studied at different levels. Among others, *C. reinhardtii* was chosen because it is one of the pioneering microorganisms used for photosynthetic therapies [22]. It is recognized as the go-to microorganism model for studying photosynthesis as it is easy to culture and has an extensive array of molecular tools available [36]. Moreover, it does not produce exotoxins or infectious agents [37], which has propelled its use as a nutritional supplement and its application in several biomedical approaches [28,29,35]. On the other hand, human melanomas were chosen as the cellular model because they tend to develop acidic and hypoxic areas [38] that lead to increased tumor malignancy [39] and high resistance to conventional ROS induction and drug treatments [40]. Among all melanoma cells available for research, the C32 human cell line was chosen due to being amelanotic, thus not interfering with the illumination settings of the study. Moreover, they have been previously combined with photosensitizers [41,42] and used for toxicological screenings of plant compounds [43,44,45,46]. Finally, due the feasibility of further illumination and administration of microalgae in prospective patients, melanomas represent a good candidate for the translation of this concept into clinics.

Previous works have described the interactions between photosynthetic microorganisms and different human cell types, such as fibroblasts [22], cardiomyocytes [25], umbilical endothelial cells [47], and various others [30]. However, to our knowledge, this is the first study that describes in detail the behavior of photosynthetic microorganisms under 3D tumor-like conditions in vitro. Compared to optimal microalgae culture settings, such conditions significantly differ in key aspects such as temperature, medium composition, and the presence of tumor cells. Surprisingly, these conditions do not seem to inhibit the intrinsic ability of the microalgae to provide significant amounts of oxygen in the presence of light. In any case, it is interesting to note that RPMI by itself did not trigger the death of microalgae; however, human physiological temperature did, as 37 °C increased cell death by two and three times in TAP and RPMI media, respectively. Unexpectedly, the presence of C32 cells significantly lowered microalgae death and increased viability, which could be due to the removal of metabolic waste or toxic byproducts from the media or the release of certain compounds that might be beneficial for microalgae metabolism, such as CO_2_. Moreover, as the protective effect was independent of light, it suggests that it is not tied to photosynthesis. This potential metabolic coupling between both cell types is interesting and deserves to be evaluated in more detail but was out of the scope of this study. Another interesting finding of this work is related to the observation that, in contrast to control conditions, after 24 h in co-culture, the metabolic activity of C32 cells did not increase, which cannot be attributed to cell death (as shown by LDH assays). Though there are no direct parallels in the literature, it has been described that colorectal cancer cell co-cultures with bacteria (*S. aureus* and *S. equisimilis*) induce a G1 arrest of the tumor cells, halting proliferation [48]. Whether these microalgae have a similar effect on tumor cell proliferation is a matter for further studies.

In terms of oxygen metabolism, even at time zero, microalgae consumption was reduced to about a third in TAP or RPMI at 37 °C. This lower consumption is contrary to what has been previously described in the literature, where oxygen consumption has been shown to increase proportionally from 28 °C to 38 °C [49]. However, in this work, the illumination setting used was different, so it cannot be directly compared to the previous studies. Concerning oxygen production, it decreases after 24-h incubation in all conditions. As this phenomenon was also observed even under control culture conditions, this decrease could be attributed to the dark pre-incubation setting and the slow kinetics of the RuBisCO enzyme [50,51]. Contrary to consumption, after 24 h of incubation, production was significantly affected by the culture media but not by temperature, which did not correlate to what was initially expected from the chlorophyll loss at 37 °C. Nevertheless, from a functional perspective, this work shows that in the presence of light microalgae were able to inhibit HIF-1α, therefore fulfilling the metabolic oxygen requirements of hypoxic melanoma cells.

The presence of *C. reinhardtii* cells affects the morphology of C32 cells, indicating possible mechanical changes and a weakening of interactions with the culture plate surface. Melanoma cells are known to easily transition into a migratory phenotype, having an overall lower expression of cadherins and integrins, as well as high expression of migration-promoting proteins when compared to non-cancerous tissue [52]. Cancer cells have also been shown to have a reduced amount of stress fibers when compared to non-cancerous tissue, leading to reduced stiffness and increased response to morphological changes. This results in a mostly amoeboid migration of the cells, though mesenchymal migration can also occur [53]. Our results suggest that, in co-culture, C32 cells show an increased presence of stress fibers in their periphery and increased fiber coherency. An increase in stress fibers suggests higher actin fiber contractility, leading to stiffer membranes. This also indicates the application of forces parallel to the adhesion surface, most likely resulting in increased mesenchymal-type migration [54]. Although the results presented here show that cells are more polarized, further experiments are needed to confirm an effective increase in migration.

With all the information described above, an in vitro 3D tumor model was engineered and characterized. Here, a collagen-GAG scaffold was chosen for cell seeding because its composition and structure resemble the tissue extracellular matrix. Moreover, this commercially available matrix has been previously shown to be biocompatible in clinical settings, as well as with several cell types in vitro [55,56,57,58], including microalgae [22] and cyanobacteria [59]. These collagen scaffolds also present unique advantages for 3D tumor cultures as they have been shown to produce hypoxic cores of cells in their center [18], a process which could be standardized for the present model. The scaffold is completely composed of type I collagen, which is found in almost all human tissues, particularly in skin [60]. On the other hand, GAGs are important for correct cell attachment and also help with scaffold water retention and compressive resistance, helping cell viability [61]. As such, this scaffold serves as a general template for the human ECM, avoiding the inherent variability and reproducibility problems of tissue-derived matrices [11]. Since this is a commercial product, it is manufactured in a controlled setting, which leads to a standardized pore size range [62].

After seeding, cells were distributed along both axes of the matrix, with larger cell density in the center of the matrix near the injection point. Melanoma cells seemed to form microtumors, which were about 100 µm in diameter and consisted of various layers of cells growing around and in between the collagen sheets of the matrix, thus closely resembling real tumor tissue. In this context, it is well known that collagen fiber directionality affects cell growth in both 2D and 3D cultures [63]. In this case, the ordered β-sheet directionality present in the scaffold results in cells aligning alongside these, which somewhat imitates a metastasis prone stroma [64]. Moreover, scaffold pores in this model serve as protected niches for microtumors and help define their shape and size, as well as affect cancer cell migration to the outside [65].

As microalgae are significantly smaller (~10 µm) than mammalian cells and the pores of the scaffold (30–120 µm [62]), they distribute homogeneously in the 3D tumor model. The LCSM and SEM images reveal that microalgae cluster inside the pores of the matrix and around the microtumors while either being in direct contact with them or in adjacent pores, confirming that this proposed model allows for the study of the interactions of both cell types in a more physiological context than standard 2D cell culture assays. Interestingly, the histological analysis showed there was significantly more cell proliferation when in the presence of the microalgae. This result was consistent in all the histological analyzed samples, but it is difficult to explain because of the short exposition time of the cells to the microalgae and the decreased metabolism observed in the co-cultures in MTT assays. Ki-67 has been described as having multiple functions during cell cycle progression [66] and is considered to be a graded marker rather than a binary one [67], which we are not able to correctly quantify with histological analysis. Thus, a more detailed cell-cycle analysis is needed to obtain more concrete results on this matter.

From a metabolic perspective, when in the presence of light, microalgae were able to produce enough oxygen to exceed the metabolic demand of C32 cells by over three times. It has been reported that mammalian tumor cells in culture have an approximate oxygen consumption range of 5 to 200 amol/(cell × s) [68], with human melanoma cells consuming an average of 37.0 ± 3.75 amol/(cell × s) when in a monolayer setting [69]. Control scaffolds consume an average of 45 pmol/s of oxygen, which can be used to roughly calculate the cell number in the scaffolds, these having roughly around 1.2 × 10^6^ cells. Similarly, each co-cultured scaffold produced an average amount of 0.22 nmol/s of oxygen, and each scaffold had around 2 × 10^7^ microalgae, meaning that the average oxygen produced per microalgae was around 11.15 amol/s. Thus, according to these rough calculations, to meet the metabolic requirements of C32 cells, a minimum ratio of 3.3 microalgae per melanoma cell should be used to avoid tumor hypoxia.

Undoubtedly, the 3D model presented here better resembles a real tumor than standard co-culture systems and, in its current shape, allows for the analysis of several specific responses to photosynthetic therapies in tumors. However, a drawback of this model is that this still represents an oversimplified system, as it lacks key accessory cell types present in primary tumors that are critical in shaping the tumor microenvironment, such as fibroblasts or endothelial cells. Nevertheless, this model could also be considered as a novel platform where, in further studies, other cell types could be incorporated. Similarly, this model does bring the extracellular matrix into context, which has been described to be of key importance for correctly modelling in vitro tumors [10]. Further, additional ECM components could also be incorporated.

Altogether, in this work, the behavior of the microalgae *C. reinhardtii* in tumor-like co-culture conditions has been studied and described. A reproducible and easy-to-use model is established for studying the potential therapeutic effects of photosynthetic therapies in tumors, particularly in relation to ROS-dependent therapies, such as photodynamic, chemo-, immune, and radiotherapy. Next, this model is also proposed to be used as a platform for studying the role of other key cell types in the tumor microenvironment, which can be included in the engineered microenvironment. Finally, the seeded 3D scaffolds can be implanted in animal models to study other key aspects of tumor behavior and photosynthesis in in vivo conditions. The findings and their implications should be discussed in the broadest context possible. Future research directions may also be highlighted.

## 4. Materials and Methods

### 4.1. Microalgae Culture

A cell-wall deficient UVM4-GFP *C. reinhardtii* strain was cultured as previously described [70]. Microalgae were grown photo-mixotrophically at room temperature (20–25 °C) on either solid Tris Acetate Phosphate (TAP) medium with 1.5% (*w*/*v*) agar or in liquid TAP medium placed in an orbital shaker (180 rpm). A lamp with white light was used to provide continuous light exposure of 30 μE/(m^2^·s) [36]. Cell density was determined using a Neubauer chamber. For all illumination settings in 2D and 3D experiments, custom LED illumination equipment (Sky-Walkers Spa., Chile) was used to illuminate the co-cultures. Light intensity and distance were set so that a constant flux of 21.47 µE/(m^2^·s) of blue light (455 nm) and 23.04 µE/(m^2^·s) of red light (630 nm) reached the cultures during all experiments.

### 4.2. Mammalian Cell Culture

The C32 human melanoma cell line was obtained from ATCC (CRL-1585). Cells were cultured in RPMI 1640 media with glutamate (Biological Industries, Cromwell, CT, USA), 10% Fetal Bovine Serum (FBS; PAN-Biotech, Aidenbach, Germany), and 1% Penicillin/Streptomycin (Biological Industries, Cromwell, CT, USA). Cells were kept in T75 flasks under standard culture conditions (37 °C and 5% CO2). Unless specified, cells were seeded and used at 70% confluence for all experiments.

### 4.3. Microalgae Morphology

Microalgae were centrifuged and resuspended in control TAP media or RPMI 1640 with 10% FBS to a final density of 1 × 10^7^ microalgae/mL. Samples were incubated for 24 h at 28 or 37 °C in darkness. For morphology analysis, microalgae samples were diluted in TAP media to a final density of 1 × 10^6^ microalgae/mL, and 10 µL was loaded onto a glass slide and taken to the light microscope. Morphology was evaluated by optical microscopy (Dmi1, Leica, Germany) with 10×/0.4 and 40×/0.65 objectives and imaged with a standard digital camera (MS60, Mshot, Guangzhou, China). Cell area quantification was performed with ImageJ software [71], where a threshold was applied to the images and the total area of each separate microalgae was calculated.

### 4.4. Microalga Viability

For flow cytometry experiments, microalgae were recovered from incubations and resuspended in 500 µL PBS to a final density of 1 × 10^6^ microalgae/mL. Samples were incubated with Sytox^®^-Green (Invitrogen, Waltham, MA, USA) cell death probe for 1 h at room temperature (RT) in darkness then centrifuged, washed twice with PBS, and resuspended in 500 µL PBS. Samples were then analyzed in a flow cytometer (FACS Canto II, BD Biosciences, Franklin Lakes, NJ, USA). All microalgae were first checked for chlorophyll fluorescence in the PE-Cy7 channel, where a gate was set up so that only chlorophyll-positive cells were analyzed. A dead microalgae control was prepared by exposing microalgae to repeated freezing and boiling cycles and then used to set the maximum fluorescence intensity and cutoff point for dead cells. Sytox^®^-Green fluorescence was quantified in the FITC channel, using the FlowJo analysis software.

### 4.5. Chlorophyll Quantification

*C. reinhardtii* microalgae were recovered from incubations, divided into aliquots of 1 × 10^7^ cells, and centrifuged at 600× *g* for 5 min. The supernatant was discarded, and the remaining pellet was frozen at −20 °C. For quantification, pellets were thawed, washed twice with 1 mL of TAP media, resuspended in 500 µL DMSO (Merck, Branchburg, NJ, USA), and left under agitation for 45 min at RT. Absorbance was measured at 665 nm and 648 nm, and chlorophyll quantification was calculated as previously described in the literature [72].

### 4.6. Microalgae Proliferation Assays

*C. reinhardtii* microalgae were recovered from incubations, washed with TAP media, and centrifuged at 600× *g* for 5 min. Pellets were plated onto sterile TAP-Agar (1.5% *w*/*v*) and left at RT for 7 days under continuous illumination in sterile conditions. Photos were taken with a standard digital camera (ILCE-6000L, Sony, Tokyo, Japan).

### 4.7. Metabolic Oxygen Profile of the Microalgae

All oxygen measurements were performed with an Oxygraph+ oxygen monitoring system (Hansatech Instruments, Pentney, UK). *C. reinhardtii* were centrifuged at 1000 rpm for 5 min and resuspended in TAP media to a final density of 1 × 10^7^ microalgae/mL. Time zero controls were created by taking microalgae directly from liquid cultures with no prior incubation and resuspending them in the appropriate media. For each sample, 1 mL was transferred to the oxygraph chamber, previously calibrated to 28 or 37 °C, and left in darkness until the oxygen concentration reached 200 nmol/mL, after which the experiment started. Samples were subjected twice to cycles of 5 min of darkness, 10 min of light, and 5 min of darkness. Slope values were calculated for each section of the curve by a linear regression.

### 4.8. Cell Co-Cultures

C32 cells were seeded onto 60 mm plates (1 × 10^6^ cells), 35 mm plates (5 × 10^5^), 12-well multiwell plates (1 × 10^5^ cells), or 96-well multiwell plates (1 × 10^4^ cells) and cultured for 24 h. Microalgae cultures were centrifuged at 1000 rpm for 5 min, resuspended in RPMI 1640 with 10% FBS, and then added to grown C32 cell cultures in a 10:1 ratio of microalgae to cells. Unless specified, all co-cultures were incubated for 24 h in standard culture conditions in darkness. When illuminated, the procedure was performed as described above (see Section 4.1).

### 4.9. Tumor Cells, Co-Culture Imaging, and Cytoskeleton Analysis

C32 cells were grown on 60 mm plates and microalgae were added to the culture, as described in Section 4.8 above. Cell morphology was analyzed by optical microscopy (Dmi1, Leica, Wetzlar, Germany) with 10×/0.4 and 40×/0.65 objectives and a standard digital camera with 200 ms of exposure and 10 gain (MS60, Mshot, Guangzhou, China). Samples were illuminated with a backlight only, and the same light intensity was kept for all samples. The total area of the plate covered by cells was calculated in ImageJ [71], where a threshold was established, and the covered area was calculated as a percentage of the total area available in each image. For fluorescence microscopy, C32 cells were grown over 12 mm glass covers until 50% confluence was reached, and microalgae were added as described above in Section 4.8. After 24 h in the co-culture, covers were washed once with PBS Ca^2+^/Mg^+^ and fixed with 4% paraformaldehyde (PFA) for 15 min at 37 °C. Fixed samples were washed three times and incubated with PBS-Triton X-100 0.1% for 10 min. Finally, samples were incubated with 1 µg/mL of Hoescht 3342 (Thermo, Waltham, MA, USA) and 0.17 µM of Phalloidin-AF546 (Thermo, Waltham, MA, USA) according to the manufacturer’s instructions. Covers were mounted on glass slides and microscopically imaged (DM500, Leica, Wetzlar, Germany) with 20×/0.4 and 40×/0.65 objectives and a standard digital camera (MS60, Mshot, Guangzhou, China). The camera setting for Hoescht detection was 200 ms of exposure and 15 gain, while exposure was set at 220 ms and 18 gain for Phalloidin-AF546. For quantifying the actin fiber intensity of the cells, images were analyzed in ImageJ [71] by using a script previously described [73]. For calculating actin fiber coherency and making the fiber directionality map, images were analyzed in ImageJ using the plugin OrientationJ according to the protocols described in the literature [74].

### 4.10. Tumor Cell Viability

For viability and cell death assays, C32 cells were grown on 96-well multiwell plates and microalgae were added as described above in Section 4.8. Here, co-cultures were incubated for 24 h in the presence or absence of light. For viability analysis, an MTT assay was performed. Cells were washed once with PBS Ca^2+^/Mg^+^ to remove the microalgae, and 10 µL of 500 µM MTT (Invitrogen, Waltham, MA, USA) and 90 µL of complete cell media were then added to each experimental point. Samples were incubated for 2 h in standard culture conditions, after which 100 µL of DMSO was added to each well. Absorbance was measured at 570/630 nm. For cell death analysis, an LDH cytotoxicity assay kit (Thermo, Waltham, MA, USA) was used. Supernatant LDH release of co-cultures and monocultures of each individual cell type was measured and calculated according to the manufacturer’s instructions.

### 4.11. Tumor Cell Immunoblotting

Cells were cultured on 35 mm culture plates and subjected to several experimental conditions. Normoxic cultures were kept under standard culture conditions, while hypoxia was induced by incubation in a sealed chamber (STEMCELL Technologies, Vancouver, BC, Canada), where a continuous nitrogen gas flow (40 mmHg/minute) was passed through for 8 min to remove oxygen. Normoxic and hypoxic cultures were incubated at 37 °C for 24 h. The cultures and the whole hypoxia chamber were illuminated as described above in Section 4.1, and samples that required darkness were protected with aluminum foil. As positive control for HIF-1α induction, 100 µM of CoCl (Merck, Branchburg, NJ, USA) was added to the culture media and left during incubation. After incubation, plates were taken to ice and homogenized in 400 µL of RIPA buffer (NaCl 150 mM, Triton X-100 1%, Sodium Deoxycholate 0.5%, SDS 0.1%, Tris-HCl pH 8.0 50 mM) containing protease and phosphatase inhibitors, before then being incubated for 40 min under constant agitation. Supernatant protein concentration was determined using a commercially available kit (Pierce BCA Protein Assay; Thermo, Waltham, MA, USA). Afterwards, samples were loaded into 7.5% SDS-PAGE gels and electrophoresis was performed. Proteins were transferred onto a PDVF membrane then blocked with TBS-Tween-20 0.2% (BSA 1%). Membranes were incubated in TBS-Tween 0.2% overnight at 4 °C with primary anti-HIF-1α (D1S7W, Cell Signaling, Danvers, MA, USA) diluted 1:1000 or anti-β-Actin (A5060, Sigma-Aldrich, St. Louis, MO, USA) diluted 1:2000, both in TBS-Tween-20 0.2%. Afterwards, membranes were incubated with a secondary Goat anti-Rabbit-HRP (Thermo, Waltham, MA, USA) diluted 1:3000 for 1 h at RT. HRP substrate (Pierce ECL WB substrate, Thermo, Waltham, MA, USA) was added to the membranes and the signal was revealed in a chemiluminescent transilluminator (myECL Imager 62236X, Thermo, Waltham, MA, USA).

### 4.12. Scaffold Seeding and Co-Cultures

Integra^®^ matrix wound dressing with a silicone layer was kindly donated by Integra Life Science and used for scaffolds (Integra LifeSciences, Princeton, NJ, USA). Six-millimeter diameter disks were cut with a punch biopsy sampler and dried with sterile gauze. Subsequently, 50 µL of cell media containing 6 × 10^5^ cells was loaded into a 1 mL disposable syringe and injected into the center of the scaffold with the silicone layer facing down. Afterwards, seeded scaffolds were placed in 12 well plates previously covered with sterile PBS-Agar 2% and left for 30 min at RT. A total of 2 mL of complete cell culture media was then carefully added to each well and replaced daily. Unless specified, tumor cells were grown for 48 h in the scaffold before being co-cultured or being used for experiments.

For seeding the microalgae into the scaffolds, 2 × 10^7^ microalgae were resuspended in 50 µL of RPMI 1640 (10% FBS). Cell-laden scaffolds were carefully dried with sterile gauze and the resuspended microalgae were added. Co-cultured scaffolds were transferred to PBS-Agar-covered multiwell plates and 2 mL of RPMI-10% FBS was carefully added to each sample. Co-cultured scaffolds were incubated for 2 or 24 h at 37 °C, according to the experimental setting.

### 4.13. Tumor Cell Viability in the Scaffold

Culture media were removed from cell-laden scaffolds and replaced with 100 µL of 500 µM MTT and 900 µL of complete cell media before then being incubated for 2 h in standard culture conditions. Optical images of scaffolds were taken on a stereomicroscope (S6D, Leica, Aidenbach, Germany) at 1.25× magnification with a standard digital camera (MS60, Mshot, Guangzhou, China). Scaffolds were kept hydrated with PBS Ca^2+^/Mg^+^ while photos were taken. Samples were then dissolved in 2 mL of DMSO with the help of a plastic tissue homogenizer. When dissolved, the remaining silicone layer was discarded, and 200 µL of each sample was transferred into a 96-multiwell plate. Absorbance was measured at 570/630 nm.

### 4.14. Tumor Cell Distribution in the Scaffold

C32 cell-laden scaffolds were grown as described in Section 4.12 above and fixed with PFA 4% for 24 h at 4 °C. For cryosection analysis, scaffolds were cut into 5 µm sections (CM1520 cryostat, Leica, Germany) and deposited onto silanized glass slides. Slides were incubated for 1 h at 50 °C and then washed in dH2O. Samples were incubated with 1 µg/mL of Hoescht 3342 (Thermo, MA, USA) in PBS for 30 min and then washed twice with PBS. Slides were analyzed with an epifluorescence microscope (Axio Observer, Zeiss, Oberkochen, Germany) using the Zeiss image software and a 20×/0.35 air objective. Tile by tile mosaic images were taken and digitally stitched. For Hoescht detection, exposure of 200 ms and 15 gain were used, while for collagen autofluorescence, exposure was set at 1.2 s and gain at 22. The nuclei fluorescence channel was isolated in ImageJ [71], a threshold was established, and total nuclei per cut were counted. From the silicone layer, the D and Q axes were defined for each cut, and each axis was subdivided into ten deciles (D) and four quartiles (Q), respectively, in Adobe Illustrator. Cells were counted within each subdivision, and a heatmap was created from the individual values of each subdivision in GraphPad Prism 8. Sections belonging to the same axis were added accordingly and averages for each separate axis were calculated and plotted. All sections in the same cut were added to obtain the total cell number, which was then compared to the original count. Shown values are expressed as a percentage of the total number of cells in a cut. Results correspond to three independent scaffolds, where 10 cuts of each were selected and counted.

### 4.15. Laser Scanning Confocal Microscopy

Cell-laden scaffolds were fixed with PFA 4% for 24 h, permeabilized with 0.2% PBS-Triton X-100 for 20 min, and washed. Scaffolds were then incubated with 1 µg/mL of Hoescht 3342 (Invitrogen, Waltham, MA, USA) and Phalloidin-AF546 (Thermo, Waltham, MA, USA) for 1 h at RT in darkness and washed twice. Samples were taken to the microscope (Airyscan, Zeiss, Oberkochen, Germany), the silicone layer was removed, and scaffolds were imaged from the top or bottom side. Images were taken with a 40×/0.65 water objective and were analyzed with Zeiss software, where they were post-processed with the Airyscan correction.

### 4.16. Scanning Electron Microscopy

Cell-laden scaffolds were fixed with glutaraldehyde 2% overnight. Samples were then dehydrated in an ethanol gradient, transferred to acetone for 1 h, and air-dried for 24 h at RT. Samples were mounted, sputtered with gold, and analyzed using 15 kV of acceleration voltage (TM3000, Hitachi, Ibaraki Japan). Images were post processed in Adobe Photoshop, where contrast was increased and cell structures were colored.

### 4.17. Histology

Cell-laden scaffolds were fixed in PFA 4%, dehydrated in ethanol, and embedded in Paraplast (Leica, Aidenbach, Germany) at 60 °C. Sections of 5 µm in thickness were cut and adhered to glass slides using 0.1% poly-L-Lysine (Sigma-Aldrich, St. Louis, MO, USA) and further dried at RT. Prior to the immunoreaction, some samples were stained with Hematoxylin/Eosin for morphological studies. Sections were deparaffinized, rehydrated, and incubated with rabbit monoclonal primary anti-ki67 (MA5-14520 Invitrogen, Waltham, MA, USA) diluted 1:50 or rabbit polyclonal anti-cleaved caspase-3 (ab2302 Abcam, Cambridge, UK) diluted 1:250, both in PBS Tween-20 0.3%, overnight at 4 °C. Non-specific staining was blocked by 30 min of immersion in Cas-Block solution (Invitrogen, Waltham, MA, USA) and goat serum (Gibco, Carlsbad, MA, USA). After extensive rinsing in PBS, all sections were incubated for 1 h at RT with HRP-conjugated goat anti-rabbit IgG (Jackson ImmunoResearch, West Grove, PA, USA) diluted 1:500 or 1:250 in PBS. The peroxidase reaction was visualized using 3,3-Diamenobenzidine chromogen from the Envision FLEX kit (Agilent, Santa Clara, CA, USA). After immunostaining, sections were slightly stained with Harris hematoxylin (Merck, Branchburg, NJ, USA). For each immunohistochemical reaction, controls were performed either by incubating the sections with PBS or by omitting the primary antibody. Subsequently, the sections were scanned at 40× magnification equivalent resolution on a whole slide scanner (Aperio Versa, Leica, Aidenbach, Germany) and images were captured with Aperio ImageScope 12.4.6 software. Images were then analyzed in ImageJ, where total nuclei per image were counted and the positive nuclei for each stain were quantified. Results are expressed as the percentage of positive nuclei per cut.

### 4.18. Scaffold Metabolic Analysis

Cell-laden scaffolds were grown and co-cultures were established as previously described (see Section 4.12). The oxygraph (see Section 4.7) was calibrated at 37 °C and 1 mL of PBS 10% FBS was introduced into the chamber. Scaffolds were transferred into the chamber and subjected to the same light–dark cycles as previously discussed (see Section 4.7). Slope values were calculated for each section of the curve by linear regression.

### 4.19. Statistical Analysis

All assays were performed in at least three independent experiments. All statistics were obtained with GraphPad Prism 8. The statistical tests used are described in each result section. The significance threshold was set at *p* ≤ 0.05.

## Figures and Tables

**Figure 1 ijms-23-13550-f001:**
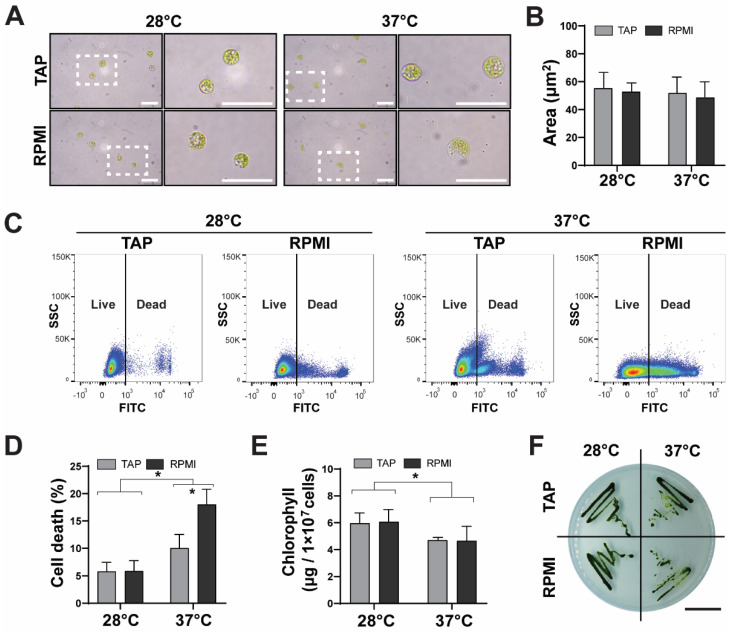
Effect of mammalian cell culture conditions on *C. reinhardtii*. Microalgae were cultured at 28 or 37 °C in TAP or RPMI for 24 h and key features were analyzed. Images of the microalgae show no major morphological changes (**A**). Cell area quantification shows no significant changes (**B**). Microalgae cell death was analyzed by FACS (**C**). Quantification shows that in both TAP and RPMI cell mortality was lower at 28 °C than 37 °C, being significantly higher for the cells cultured at 37 °C in RPMI (**D**). Chlorophyll content significantly decreased at 37 °C, independent of medium (**E**). When recovered, microalgae were able to proliferate in agar plates in all conditions (**F**). Scale bars represent 20 µm in (**A**) and 1 mm in (**F**). All assays were performed in at least three independent experiments and data are presented as average + standard error (SE). *p* ≤ 0.05 was considered as significant using a two-way ANOVA, with different letters implying statistical differences. * = *p* ≤ 0.05.

**Figure 2 ijms-23-13550-f002:**
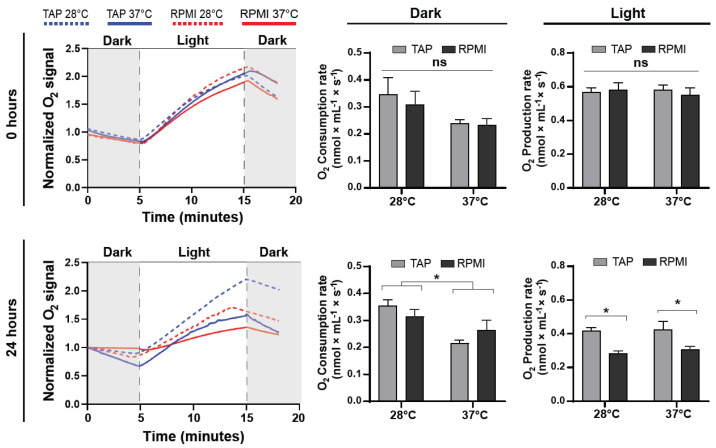
Effect of the mammalian culture conditions in the metabolic profile of *C. reinhardtii*. Microalgae were cultured at 28 or 37 °C in TAP or RPMI for 24 h, and their oxygen production and consumption were evaluated. At time zero, no significant differences were observed among all groups (upper panels). After 24 h in culture, oxygen consumption rate decreased at 37 °C, while production decreased in RPMI at both temperatures. All assays were performed in at least three independent experiments and data are presented as average + SE. *p* ≤ 0.05 was considered as significant using a two-way ANOVA, with different letters implying statistical differences. * = *p* ≤ 0.05, ns = not significant.

**Figure 3 ijms-23-13550-f003:**
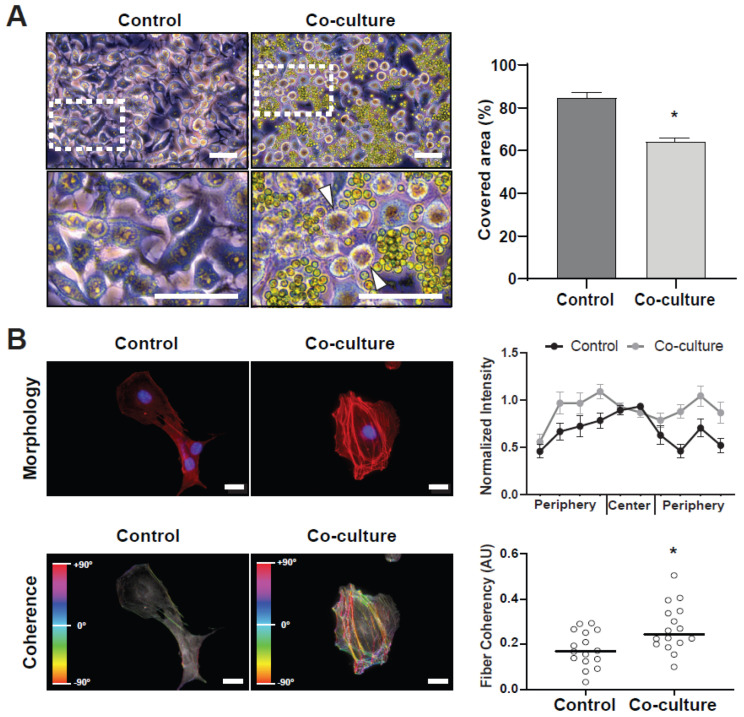
Co-cultures of *C. reinhardtii* and C32 tumor cells. Microalgae were added to sub-confluent C32 seeded plates and left for 24 h in standard culture conditions. A general view of the co-culture is shown in (**A**, **left**), with arrowheads showing cells that are detaching from the plate. A significantly lower surface area was found to be covered by C32 cells in co-culture conditions (**A**, **right**). A cytoskeleton analysis of the cells shows that actin fibers increased in the periphery of the cells when co-cultured (**B**, **upper right**). The actin fiber directionality analysis is color coded, and results show significantly increased fiber coherency in co-culture conditions (**B**, **lower right**). All assays were performed in at least three independent experiments and data are presented as average + SE. Scale bars represent 50 µm in (**A**) and 20 µm in (**B**). *p* ≤ 0.05 was considered as significant using a Mann–Whitney test. * = *p* ≤ 0.05.

**Figure 4 ijms-23-13550-f004:**
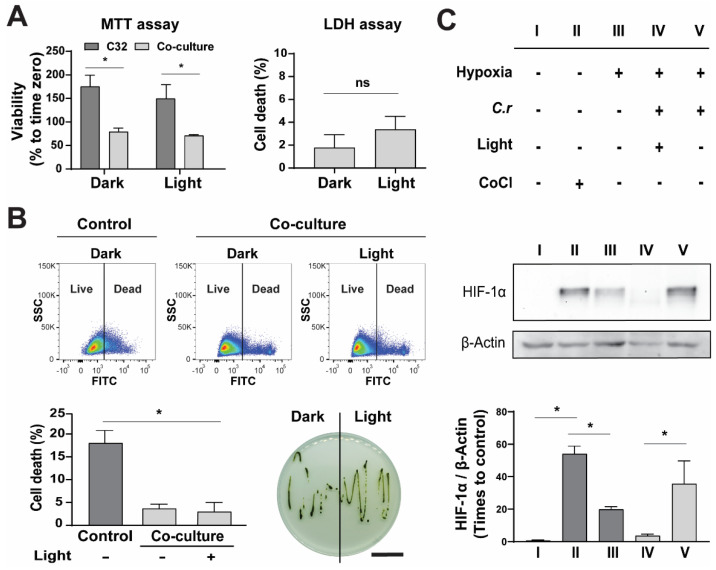
Effect of illumination in co-cultures of *C. reinhardtii* and C32 tumor cells. Cells were co-cultured for 24 h in the absence or presence of light. As shown by an MTT assay, regardless of the presence of light, the mitochondrial activity of C32 cells decreased significantly in co-cultures (**A**, **left**), while cell death did not vary amongst groups but was slightly increased from control (**A**, **right**). Similarly, light did not affect viability nor proliferation capacity of the microalgae in co-cultures; however, its mortality was significantly lower when compared to control monocultures in cell media (**B**). The effect of light on hypoxia was evaluated (**C**) by HIF-1α expression. Cells were grown in normoxia (I), incubated with CoCl (II) or grown in hypoxia (III). HIF-1α decreased in illuminated hypoxic co-cultures (IV) when compared to hypoxic co-cultures in the dark (V). All assays were performed in at least three independent experiments and data are presented as average + SE. The scale bar represents 1 mm in (**B**). *p* ≤ 0.05 was considered as significant by using a two-way ANOVA test in an MTT viability assay, a Mann–Whitney test in an LDH release assay, and one-way ANOVA in B and C. * = *p* ≤ 0.05, ns = not significant.

**Figure 5 ijms-23-13550-f005:**
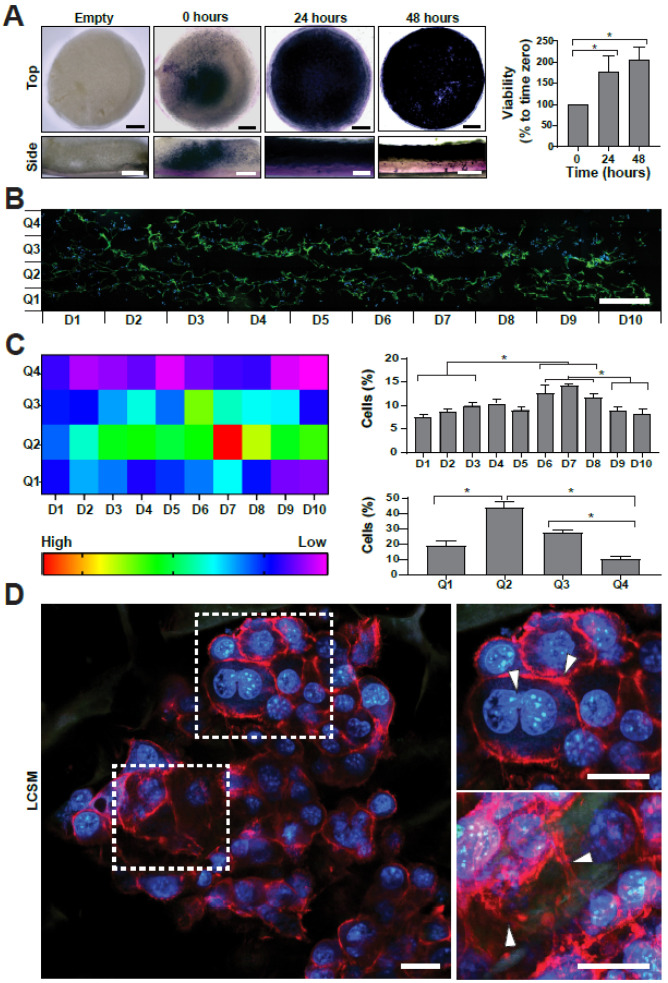
Establishment of a 3D tumor model. C32 cells were seeded in a collagen-GAG scaffold; 48 h post seeding, a significant increase in metabolic activity was quantified by MTT assays (**A**). 3D tumor model was cut into 5 µm slides and cell distribution was visualized by fluorescence microscopy (**B**). A heatmap (**C**, **upper left**) shows the cell distribution in the scaffold, showing a non-homogeneous internal distribution. Detailed analysis of the sections (**C**, **right**) shows that cells concentrate in the central sections of the scaffold. LCSM analysis of the seeded scaffolds shows tumor-like structures (**D**, **left**). Arrowheads in the magnified areas show cell–cell interactions and cell proliferation (**D**, **upper right**) and cell–matrix interactions (**D**, **lower right**). All assays were performed in at least three independent experiments and data are presented as average + SE. Scale bars represent 1 mm in (**A**,**B**) and 25 µm in (**D**). *p* ≤ 0.05 was considered as significant using a one-way ANOVA test. * = *p* ≤ 0.05.

**Figure 6 ijms-23-13550-f006:**
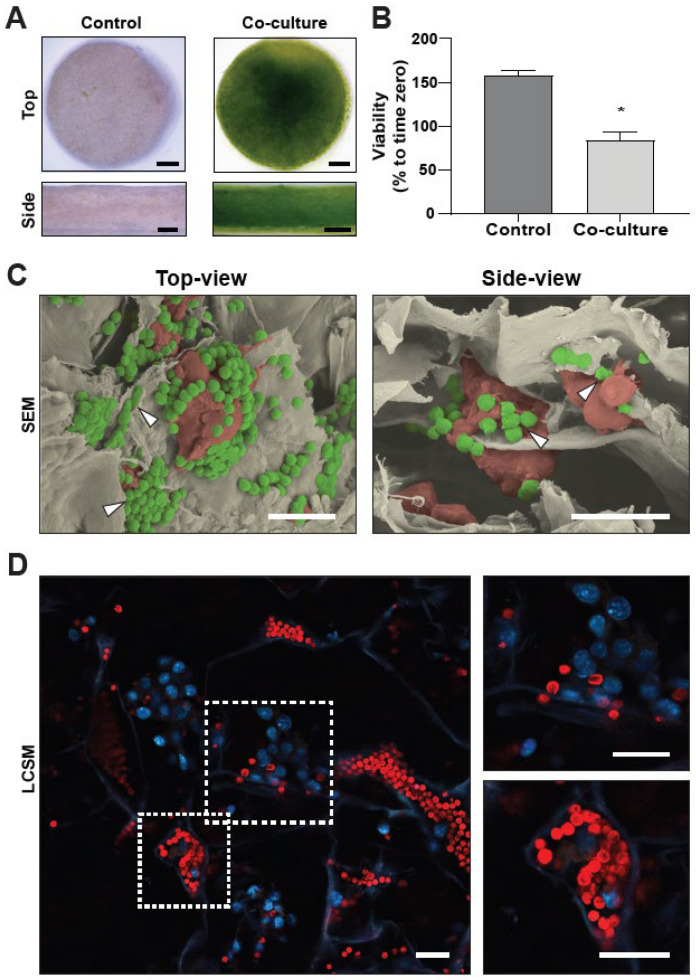
Morphological analysis of a photosynthetic 3D tumor model. C32 cells were seeded in a collagen-GAG scaffold and cultured for 48 h, with microalgae subsequently added. Overview images of the scaffolds show a generally homogeneous distribution of the microalgae (**A**). MTT analysis shows significantly lower metabolic activity in co-cultured scaffolds compared to control scaffolds (**B**). SEM images of 2-h co-cultured scaffolds show tumor cells in direct contact with microalgae (**C**). The top-view arrowheads (**C**, **left**) show microalgae in between collagen sheets, while side-view arrowheads (**C**, **right**) indicate microalgae interacting with cancer cells. Detailed LCSM images of 2-h co-cultured matrices (**D**) show direct physical contact between C32 cells (DAPI/blue) and microalgae (chlorophyll/red) (**D**, **left**). Right panels show magnifications of the indicated sections on the left panel. All assays were performed in at least three independent experiments and data are presented as average + SE. Scale bars represent 1 mm for (**A**), 20 µm for (**C**), and 50 µm for (**D**). *p* ≤ 0.05 was considered as significant using a Mann–Whitney test. * = *p* ≤ 0.05.

**Figure 7 ijms-23-13550-f007:**
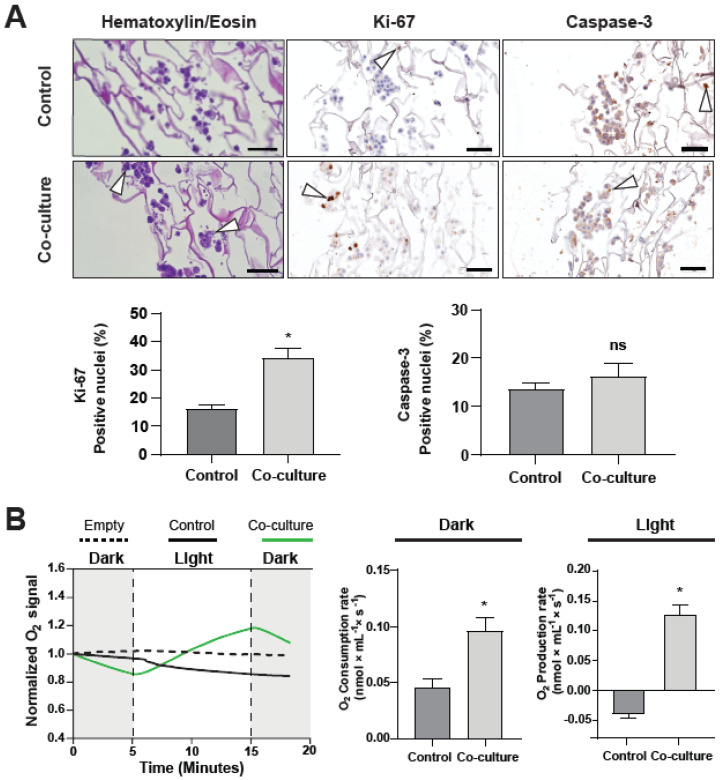
Characterization of a photosynthetic tumor model. Microalgae were seeded into scaffolds with grown tumors and incubated for two hours under physiological conditions. A histologic analysis of the scaffolds was conducted by staining tumor cells with Hematoxylin/Eosin against Ki-67 and Caspase-3. Arrowheads indicate the presence of microalgae clusters (H/E) and positive nuclei for the corresponding stain (Ki-67 and Caspase-3). Ki-67 quantification (**A**, **lower left**) shows a significant increase in tumor cell proliferation when microalgae are present, while Caspase-3 quantification (**A**, **lower right**) shows no significant differences were detected. Oxygraphic studies of co-cultured scaffolds show that, in darkness (**B**, **middle**), co-cultured scaffolds consume twice as much oxygen as control scaffolds. Under light (**B**, **right**), control (C32) scaffolds keep consuming oxygen, while co-cultured scaffolds produce over twice the consumed amount of oxygen. All assays were performed in at least three independent experiments and data are presented as average + SE. Scale bars in (**A**) represent 100 µm. In all experiments, *p* ≤ 0.05 was considered as significant using a Mann–Whitney test. * = *p* ≤ 0.05, ns = not significant.

## Data Availability

All data associated with this study are presented in this paper and can be shared with approved outside collaborators under a materials transfer agreement; requests should be sent to J.T.E., jte@uc.cl.

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
