# Peer review of "Towards an In Vitro 3D Model for Photosynthetic Cancer Treatment: A Study of Microalgae and Tumor Cell Interactions"

_ijms, 2022, doi:10.3390/ijms232113550_

Round 1

Reviewer 1 Report

This study developed a 3D tumor spheroid model together with photosynthetic microorganisms, which could oxygenate the tumor microenvironment. The manuscript was well organized and presented. Several comments are listed below.

1. In Fig. 7, after encapsulating tumor cells and microorganisms in collagen-GAG scaffold, how did the authors illuminate the microorganisms in a 3D structure? whether the gel scaffold was transparent?

2. For a tumor spheroid with several mm size, the low oxygen level or high hypoxic level would be expected in the centre but not periphery. In Fig. 5-7, the authors should present the results to show the oxygen level or hypoxic level, cell death, and proliferation in different positions of the tumor spheroid (periphery vs center) with and without microorganisms in the presence and absence of light, which are critical to show the influence of photosynthetic microorganisms on hypoxia and cellular functions in the 3D spheroid model.

3. Why did the death of microorganisms decrease significantly after co-culture with tumor cells in Fig. 4B?

4. For all the figures, the p-values or "*" should be directly added to replace "a", "b". It is very confusing in the current format. 

5. There are many grammars in this manuscript: such as figure captions.

Fig. 1 "Effect of mammalian cell culture conditions in C. reinhardtii." should be "Effect of mammalian cell culture conditions on C. reinhardtii."

Reviewer 2 Report

The manuscript by J.T. Egaña and co-workers describes the interactions of tumour cells with one microalgae. As some treatments require oxygenated tumours to be efficient, it is essential to bring enough oxygen at the tumour site and using photosynthesis is a way to achieve this. The manuscript goes through different positive and negative controls to ensure the tumour cells and the algae can be co-cultured before engineering a 3D model that mimics the tumour microenvironment and that contains the algae and study the interactions between the algae and the tumour cells.

The topic is very interesting and raises questions regarding the use of living organisms in human.

I have noticed only two typos:

- line 419 should read "resulting in a mostly amoeboid-type ..."

- line 740 should read "JTE who owns ..."

The references are not all formated using the same template (pages + DOI, pages only, DOI only, pages or DOI missing).

Reviewer 3 Report

the authors characterized co-culture of the microalga C. reinhardtii and C32 melanoma cells, with the potential to develop into a 3D model to evaluate photosynthetic treatment for cancer. overall the results solid, and the conclusion convincing.

my only concern is language. it need polishing, especially the Discussion. Just a few confusing examples:

Line 360: As well as being the go-to model microorganism for studying photosynthesis, as it is both simple to culture and it has an extensive array of molecular tools are available.

Line 419: ...resulting in a mostly an amoeboid-type migration

Line 424: Although OUR results show...Add our 

Line 453: these distribute homogeneously in the 3D tumor model. Change these to they, or spell out.

Line 482: as a novel platform were, ...Change were to where

Line 488: ...in tumor-like co-culture conditions in studied and described. Change in to is. 

Line 494-497: it seems two sentences were sticked together without proper punctuation. 

Round 2

Reviewer 1 Report

The authors addressed all my concerns.